# Role of Regulatory T Cells in Pulmonary Ageing and COPD Development

**DOI:** 10.3390/ijms26083721

**Published:** 2025-04-15

**Authors:** Virginija Šileikienė, Laimutė Jurgauskienė

**Affiliations:** 1Clinic of Chest Diseases, Immunology and Allergology, Faculty of Medicine, Institute of Clinical Medicine, Vilnius University, LT-03101 Vilnius, Lithuania; 2Clinic of Cardiac and Vascular Diseases, Faculty of Medicine, Vilnius University, LT-03101 Vilnius, Lithuania; laimute.jurgauskiene@santa.lt

**Keywords:** Tregs, COPD pathogenesis, pulmonary ageing

## Abstract

Chronic obstructive pulmonary disease (COPD) is recognized as a long-term inflammatory lung condition, predominantly resulting from smoking tobacco. While all smokers exhibit some level of pulmonary inflammation, only about 15–20% go on to develop significant COPD, indicating that specific individual factors may enhance these inflammatory responses and contribute to the disease’s progression. T regulatory cell (Treg) activity is crucial in mediating pulmonary inflammation in COPD. With accumulating evidence supporting the autoimmune characteristics of COPD, there has been an increasing focus on the role Treg cells play in the disease’s initiation and development. This article aims to review the existing literature regarding Treg cells and their influence on COPD pathogenesis and lung ageing. Treg-mediated suppression is a critical mechanism in the negative regulation of immune-related inflammation, which is significant in various disorders, including autoimmunity, allergies, infections (both acute and chronic), and cancer. The lungs of ageing individuals often resemble those affected by COPD, leading to the perception of COPD as a condition that accelerates lung ageing. Changes in Treg cells with age correspond to decreased adaptive immune responses and a higher likelihood of immune-related disorders. The increased presence of Treg cells in older adults may help explain several immunological conditions commonly associated with ageing, which include malignancies, infections, and COPD.

## 1. Introduction

Chronic obstructive pulmonary disease (COPD) is a persistent inflammatory condition impacting the lungs that leads to not fully reversible airflow limitation. The World Health Organization projects that COPD will continue to be the third leading cause of death until 2030 [1]. While genetic and environmental factors contribute to COPD, smoking is the primary cause in developed nations [2]. Notably, even though smokers generally experience a pulmonary inflammatory response, only 15–20% develop significant COPD, highlighting that individual genetic and intrinsic factors may intensify these responses and influence disease advancement [3]. This brings forth a crucial question: What mechanisms enable most smokers to avoid developing COPD? The progression of COPD can vary widely among individuals; some people may maintain a stable condition, while others face a relentless decline, resulting in severe breathlessness. The factors that determine the initial course and severity of COPD in smokers remain poorly understood.

The development of COPD is made up of and marked by various mechanisms that interact and coexist [4,5]. T cell-mediated adaptive immunity is significant in regulating airway inflammation throughout the advancement of COPD. An adaptive immune reaction can arise from either self-antigens [6] or external antigens (such as bacterial, viral, or fungal agents). It can also play a role in the pathogenesis of COPD [7,8].

This review evaluates existing research regarding the influence of autoimmunity and regulatory T (Treg) cells on the pathogenesis of COPD, specifically analyzing studies on Treg cells in COPD patients’ lungs and blood.

## 2. Treg Cells

Immune homeostasis is a unique form of immunological tolerance orchestrated by a group of CD4+ T cells, known as regulatory T (Treg) cells, exhibiting strong immunosuppressive capabilities. Treg cells are a specific subset of CD4+ T lymphocytes characterized by the expression of the transcription factor forkhead-box-P3 (FoxP3), abundant levels of the interleukin (IL-2) receptor α-chain (CD25), and play a crucial role in maintaining tolerance to self-antigens, while preventing autoimmune reactions [9].

The suppression mediated by Treg cells is a crucial mechanism for the negative regulation of immune-mediated inflammation, playing a significant role in autoimmune and auto-inflammatory diseases, allergies, acute and chronic infections, and cancer.

Mature Treg cells originating from the thymus migrate into peripheral tissues and are essential for inhibiting inappropriate immune responses triggered by self-reactive T cells. In peripheral tissues, CD4+Foxp3 T cells can gain Foxp3 expression and differentiate into peripherally induced Treg (pTreg) cells when stimulated through their T cell receptor (TCR) in the presence of specific environmental signals, such as transforming growth factor beta (TGF-β) and additional co-stimulatory signals [2].

Moreover, thymus-derived Treg (tTreg) cells located in secondary lymphoid organs, referred to as central Treg (cTreg) cells, travel through the circulation and transform into effector Treg (eTreg) cells, depending on TCR activation. Upon activation, Foxp3+ regulatory T (Treg) cells differentiate into eTreg cells, essential for sustaining peripheral immune homeostasis and tolerance. During this differentiation process, they downregulate specific molecules like CD62L and CCR7, while upregulating CD44, chemokine receptors, and integrins that facilitate their movement into non-lymphoid tissues [3,4].

Grossman W et al. proposed that Treg cells likely exert their suppressive effects through various mechanisms. Research has demonstrated that activated regulatory T cells have the ability to eliminate effector cells or antigen-presenting cells by releasing perforin and granzymes [10]. Tregs may interact with APC functionally via co-stimulatory molecules (CD80/CD86, CTLA-4) expressed on the surface of dendritic cells (DCs) ([11], Figure 1). In addition, soluble factors, such as IL-10, IL-35, TGF-β, and LAG3, also modulate Tregs functions [12,13]. Regulatory T cells expressing FoxP3 and ICOS can inhibit T cells and dendritic cells by releasing cytokines TGF-β and IL-10, respectively [14]. Furthermore, Tregs expressing HLA-DR molecules can stimulate early contact-dependent inhibition [15]. Akkaya et al. [16] demonstrated that antigen-specific Tregs can downregulate the antigen-presenting capabilities of DCs by depleting peptide–MHC II on their surface. This process is noteworthy for its antigen specificity and represents a novel mechanism in regard to Tregs’ suppressive function.

It is known that Tregs express other inhibitory molecules, such as CD73, CD39, T cell immunoglobulin, and ITIM domains (TIGIT) [17]. Tregs induce senescence in both naïve T cells (Tn) and effector t cells (Tef), resulting in alterations to the interleukin profile and phenotype of Tef cells, making them highly suppressive. Tregs regulate the senescence of T cell responders by modulating various signalling pathways, including ERK1/2 and p38, as well as cell cycle regulatory elements, such as p21, p16, and p53 [18].

Over time, it has become increasingly clear that Treg cells are diverse. More than 40 distinct Treg subpopulations, characterized by different surface markers, have been identified [19,20]. The most widely accepted classification method, proposed by Miyara et al. [21], divides human Treg cells into three functionally and phenotypically distinct subpopulations: CD45RA+Foxp3low resting Tregs (r Tregs, fraction I), CD45RA−Foxp3hi activated Tregs (a Tregs, fraction II), both of which exhibit suppressive properties in vitro, and CD45RA−Foxp3lo pro-inflammatory cytokine-secreting cells (fraction III), which lack suppressive functions.

Identifying specific markers to define Treg cells and differentiate them from activated effector T cells is critical for fully understanding their function. However, a consensus on reliable specific markers for identifying Treg cells has yet to be achieved. It has also been indicated that Foxp3 may not be a definitive marker for Tregs, as its expression can be transiently elevated in activated effector T cells [22], and Treg cells can lose Foxp3 expression and convert into effector T cells [23]. Along similar lines, the search for definitive identifiers continues. At the same time, various specific markers for Tregs exist, such as the glucocorticoid-induced tumor necrosis factor receptor and CD25, adhesion molecule CD62L, PD-1, cytotoxic T lymphocyte antigen-4 (CTLA-4), and Helios, are also upregulated upon activation [24,25]. CD127, a surface marker used to isolate genuine human Treg cells via flow cytometry, is not a specific marker [26].

## 3. Treg Cells and COPD

### 3.1. COPD Exhibits Features of Autoimmunity

The involvement of autoimmune-related inflammation in COPD was initially suggested by Saetta et al., when histological examinations of human lung tissue revealed a predominance of CD8+ T cells in airway biopsies from individuals with COPD who had a history of smoking [27]. Over the past few decades, compelling clinical and pathological findings have led to a novel understanding: in individuals predisposed to disease, exposure to cigarette smoke may initiate enduring memory T cell inflammatory responses that persist long after the exposure has ceased. Crucially, the ongoing recruitment of activated lung antigen-presenting cells (APCs) can further promote these auto-inflammatory T cell responses, contributing to chronic and progressive lung damage [28].

Autoimmunity is now increasingly recognized as a crucial factor in the development of COPD. Studies have indicated that regulatory T (Treg) cells play a significant role in modulating autoimmunity. Domagala-Kulawik et al. [29] found that patients with mild-to-moderate COPD exhibited a notably lower proportion of peripheral blood CD4+ CD25+ lymphocytes than healthy individuals. This highlighted that COPD patients have reduced levels of these immune cells in their blood, suggesting an alteration in their immune response relative to healthy individuals. Concerning the anti-inflammatory role of Treg responses, patients with severe and very severe COPD had fewer Treg cells in their bronchial epithelium than those with mild or moderate disease and healthy smokers [30]. Furthermore, Chu et al. reported a reduction in Foxp3+ cells and gene and protein expression in tissue samples from moderate and severe COPD patients compared to healthy smokers and non-smokers [31].

More recently, Ström et al. examined the ratio of Tregs with regulatory capabilities (Foxp3+/CD4+CD25^bright^) in COPD smokers, alongside healthy smokers and control groups [32]. Although they did not find significant differences across the groups, it was noted that COPD patients experiencing a rapid decline in lung function had significantly fewer regulatory Tregs than those with a slower decline, suggesting that a failure to regulate the inflammatory response after smoking might contribute to a faster reduction in FEV_1_.

An intriguing finding by Xiang-Nan Li and colleagues reported discrepancies in the proportions of Th17 and Treg cells and their associated cytokines in COPD patients [33]. The levels of the pro-inflammatory Th17 cells were significantly elevated during acute exacerbations of COPD compared to stable COPD patients and smokers with normal lung function. Conversely, Treg cell levels were lower than in stable COPD but higher than in healthy controls, indicating a shift in COPD patients’ Th17/Treg cell balance. Unlike Th17 cells, Treg cells can exert immunosuppressive effects by secreting anti-inflammatory cytokines such as interleukin (IL)-10 and transforming growth factor (TGF)-β1, which promote immune tolerance and counteract inflammation.

Recently, new markers for identifying Tregs have emerged, indicating a need for better or more effective markers to recognize Tregs in human subjects accurately. One such marker, CD127, was shown to have an inverse relationship with FoxP3 and the suppressive functions of human CD4+ Tregs in peripheral blood [34]. Additionally, Xia Yang and colleagues [35] discovered that individuals with stable COPD had notably higher proportions of CD4+CD25+CD45RO+ Tregs, but lower counts of CD4+CD25+CD127^low^ Tregs, CD4+CD25+CD45RA+ Tregs, and CD4+CD25+CD62L+ Tregs, suggesting a shift towards immune tolerance in patients with COPD due to an imbalance between immunosuppressive and immunoenhancing CD4+CD25+ T cell subsets.

A potential role for CD4+CD25+^bright^ T cells in the development of COPD was proposed in a study by Chiapori et al. [36], which indicated that the percentage of these Tregs was significantly reduced in current and former COPD patients that smoked compared to volunteers. However, this study did not categorize T regulatory cells based on disease severity, focusing only on current or past smoking status.

In the study by Barcelo et al. [37], it was noted that CD4+ CD25+^bright^ cells were significantly more abundant in bronchoalveolar lavage fluid from smokers with normal lung function compared to those who never smoked and patients with moderate COPD. Another study also found that bronchoalveolar lavage samples from smokers and COPD patients had higher levels of CD4+CD25+^bright^ cells compared to healthy individuals who had never smoked [38].

From these data, it is clear that Treg cells play a significant role in the pathogenesis of COPD as an anti-inflammatory regulator. In the case of COPD, especially in more severe stages, there is a dysfunction of this immune mechanism.

### 3.2. A Distinct Phenotype of COPD May Involve Alternate Immunologic Mechanisms

Patients exhibiting more pronounced emphysema at the baseline showed a reduced forced expiratory volume, a lower body mass index (BMI), compromised functional capacity, and an increased incidence of osteoporosis, thereby resembling the multiple organ loss of tissue (MOLT) phenotype. The presence of glycoprotein A dominant repeat (GARP) has been observed exclusively in activated human naturally occurring regulatory T cells and their clones rather than in activated effector T cells, highlighting GARP as a marker for authentic Tregs [39]. This study indicates that a reduction in highly suppressive Tregs, characterized by the GARP+ subset, alongside an accumulation of pro-inflammatory cytokine-producing Tregs defined by the GARP− subset, is associated with more advanced stages of the MOLT phenotype in COPD.

### 3.3. Tregs in COPD Tissue

There is an ongoing discussion surrounding the quantity of Tregs in COPD-affected tissues. Some research suggests that stable COPD patients’ levels of CD4+ CD25+ FoxP3 Tregs in bronchial biopsies [40] or lung tissue [41,42] do not significantly differ from those of healthy individuals. However, they are reduced in the small airways of COPD patients, correlating negatively with the severity of airflow obstruction [43,44]. Conversely, our previous research [30] found significantly fewer intraepithelial CD4+ CD25+ lymphocytes in patients with severe/very severe COPD (GOLD III–IV) and among the control non-smokers compared to mild/moderate COPD (GOLD I–II) and control smokers. This implies that severe COPD develops in individuals with reduced levels of Tregs in both blood and respiratory tissue. At the same time, higher concentrations of Tregs in smokers without COPD suggest a potential protective role of Tregs against the development of COPD. The studies referenced analyzed lung tissue from patients undergoing surgical removal for carcinoma, which could affect the findings due to the use of neoplastic processes [41,43,44]. Our data [30] were unaffected by cancer, as neither our COPD patients nor the controls exhibited any oncological maladies. Various studies suggest that the function of T regulatory cells in modulating the immune response may differ across lung regions and among various COPD clinical studies.

Given the existence of B cell lymphoid follicles in patients with advanced COPD and the identification of various autoantibodies in certain COPD patients, COPD has been classified as an autoimmune disorder [45]. Plumb and colleagues [46] were pioneers in examining Tregs within pulmonary lymphoid follicles. Their findings indicated a heightened proportion of Treg cells in the pulmonary lymphoid follicles of patients with moderate severity COPD compared to smokers and non-smokers. However, no significant difference was noted in Treg cell proportions between the groups within clusters and the subepithelium. Despite the elevated number of Treg cells in COPD, the authors posited that the regulatory functions of Tregs could be disrupted within COPD lymphoid follicles. It is noteworthy that their study focused solely on individuals with moderate COPD. This detail holds significance as the prevalence of lymphoid follicles markedly increases from 25% to 30% in severe and very severe COPD for reasons that remain unclear [47]. The transition from moderate-to-severe COPD, associated with a rapid rise in follicle numbers, may result from a breakdown in regulatory mechanisms meant to suppress autoimmunity.

D’Alessio and colleagues demonstrated that Tregs can provide a protective effect against lung injury: Tregs characterized as CD4+, CD25+, and FoxP3+ facilitate the resolution of lung injuries in experimental mouse models following the administration of intratracheal lipopolysaccharide. These Tregs are also present in the bronchoalveolar lavage fluid of patients experiencing acute lung injury, indicating that Treg cells may influence innate immune responses during the recovery phase from lung injury [48]. Table 1 summarizes the primary findings regarding Tregs and COPD from various clinical studies.

### 3.4. Metabolism and Immune Response in COPD

The impact of leptin on Tregs is complex, and there are conflicting results in regard to different diseases and differing murine models. A recent investigation by Italian researchers [56] highlighted the interplay between metabolism and immune response in the development and progression of COPD. Patients with varying stages of COPD exhibited a gradual rise in systemic leptin. Leptin is known to act in two ways: as a hormone and as a cytokine. As a cytokine, leptin acts in a pro-inflammatory manner. High leptin levels reduce Treg levels, which leads to impaired immunoregulation in COPD, with effector T cells predominating, promoting inflammation and worsening lung function. By examining the molecular level interactions, Bruzzaniti et al. [56] found that leptin inhibited the expression of forkhead-boxP3 (FoxP3) and its splicing variants containing the exon 2 (FoxP3-E2) that correlated inversely with inflammation and weakened lung function during COPD progression. These researchers found that the immunometabolic pathomechanism leading to COPD progression is characterized by leptin overproduction, a decline in the expression of FoxP3 splicing forms, and an impairment in Treg cell generation and function. They examined circulating peripheral Treg (pTreg) cells in groups of healthy non-smokers, healthy smokers, and COPD patients, across various disease stages. The researchers used two specific FoxP3 monoclonal antibodies to stain pTreg cells: one that identifies all splicing variants of the FoxP3 protein (designated as CD4+FoxP3-all+ Tregs) and another that targets variants containing exon 2 (CD4+FoxP3-E2 Tregs). The latter variant has been recently recognised as crucial for conferring suppressive capabilities to Treg cells. They found that COPD patients classified as GOLD stage II had a higher count of CD4+FoxP3-all+ and CD4+FoxP3-E2+ pTreg lymphocytes than healthy individuals. However, the prevalence of both CD4+FoxP3+ Treg cell subsets declined significantly in patients with COPD at GOLD stages III and IV. Clinically, the levels of CD4+FoxP3-all+ and CD4+FoxP3-E2+ pTreg cells in the blood of COPD patients correlated positively with lung function. It remains uncertain which induced FoxP3+ Treg cells promote systemic self-tolerance, and there is still debate regarding the specific cell markers that can be used to separate Treg cells for independent study of their properties.

### 3.5. Granzyme B as an Effector Molecule and Potential Functional Marker for Treg Cells in COPD

Gondek DC et al. [57] noted that the perforin–granzyme pathway is essential for the functions of NK and CD8+ T cells and can also be utilized by Treg cells to modulate their activity. Consequently, the granzyme pathway is one of the methods through which Treg cells manage immunological responses. Treg cells are hypothesized to regulate these responses through the granzyme B (GzmB) pathway. A recent study by Kim et al. [58] indicated that immunohistochemical analysis of GzmB in surgically removed lungs exhibiting centrilobular emphysema demonstrated that the correlation between the quantity of GzmB+ cells and FEV1% was similar to the association between the Treg cell count and FEV1% in COPD lungs, suggesting that GzmB could serve as a functional marker for Treg cells. The fraction of GzmB+ cells in small airways, the number of alveolar GzmB+ cells, and GzmB levels, measured using an enzyme-linked immunosorbent assay, in the lung tissues of smokers, were significantly associated with FEV1%. These authors speculate that the presence of GzmB in lung tissue may influence COPD progression by acting as an effector molecule to modulate inflammation. Therefore, interventions aimed at enhancing the presence of GzmB-producing immunosuppressive cells during the early stages of COPD might help to prevent or slow down the disease’s progression. While this latest publication is intriguing, it is highly speculative and presents no direct evidence to support its thesis.

### 3.6. Does the Level of Treg Cells Influence the Development of Lung Cancer?

A distinct relationship between lung cancer and COPD has been observed. A recent study by Wauters et al. [59] revealed specific mutation patterns and molecular characteristics in non-small-cell lung cancer (NSCLC) that increase in the context of COPD. These findings have demonstrated for the first time that lung tumors in COPD patients differ from those in non-COPD individuals due to a unique tumor microenvironment characterized by a reduced number of CD4+ Treg cells. The experimental results indicate that the heightened methylation status of immune-related genes in COPD-associated cancers corresponds with a lower infiltration of CD3+ and CD4+ immune cells in the surrounding tumor stroma. This is seemingly at odds with T cell expression in patients with COPD, where T cell markers typically show overexpression. However, upon examining various Th cell subtypes (Th1, Th2, Th9, Th17, Th22, Treg, and follicular T cells), researchers found that the decrease in the CD4+ T cell component was primarily due to a diminished Treg response. CD4+CD25+FoxP3 regulatory T cells play a crucial role in suppressing excessive T cell responses, serving as a significant mechanism for maintaining tolerance. A shortage of Treg cells has been identified in the lungs of COPD patients, which can disrupt the immune system’s ability to tolerate autoantigens, potentially leading to immune-mediated lung damage [6]. This observation is crucial, with significant clinical implications for improving COPD and lung cancer [60].

## 4. Treg Cells and Pulmonary Ageing

Ageing lungs exhibit many similarities to those affected by COPD, and COPD is often regarded as a condition of accelerated lung ageing. The incidence of COPD is significantly higher among individuals over 60 compared to younger age groups [61]. The impact of ageing on the immune system, called immune senescence, affects both innate and adaptive immune responses and is recognized as a key factor in the onset of COPD [62]. Age-related changes are observed in various immune cells and cytokine production, cell apoptosis, TCR diversity, and other cellular functions.

The thymus progressively deteriorates with ageing, leading to a diminished capacity for T cell production, including Tregs [63,64]. Nevertheless, the overall prevalence of Tregs in the periphery remains stable due to increased peripheral generation of Tregs, compensating for the gradual loss of thymic function. As a result, the number of naïve Tregs declines, while the number of memory Tregs and induced Tregs increase with age [65]. These modifications in Tregs during ageing contribute to reduced adaptive immune responses and a greater risk of immune-mediated disorders [66]. The gradual decline in the immune system with age hampers responses to new antigens, encourages the development of autoimmunity, and heightens susceptibility to infections and potential shifts in the airway microbiome [5]. The accumulation of Treg cells in older individuals helps explain some age-associated immunological disorders, including malignancies, infections, and COPD [67]. Beyond hindering adaptive immune mechanisms, ageing also influences the innate immune system, resulting in persistently elevated basal levels of systemic inflammation marked by higher levels of pro-inflammatory cytokines, such as IL-1, IL-6, and TNF-α. This phenomenon, called “inflammaging” [61,62], highlights the interplay between senescence and chronic inflammation.

Given the physiological and immunological similarities between chronic obstructive pulmonary disease (COPD) and ageing, COPD has been regarded as an “accelerated ageing phenotype”. Both ageing and COPD share common molecular mechanisms, such as increased activation of NF-κB, oxidative stress, telomere shortening, and impaired DNA repair. These factors lead to significant dysregulation of the immune system [68].

Ageing is associated with decreased epithelial barrier function [69], abnormalities in cilia structure and function [70], and reduced production of antimicrobial and anti-inflammatory peptides by epithelial cells, including the secretory leukocyte protease inhibitor (SLPI) [71]. Unsurprisingly, both alveolar and airway epithelial cells in smokers with COPD have been reported to show increased numbers of senescent cells compared to healthy controls [72]. Excessive cellular senescence is a pathogenic mechanism in COPD, highlighting the potential of senolytic therapies to reduce the burden of senescent cells in various human diseases [73].

Our literature review suggests a framework for understanding COPD development, highlighting the essential roles of regulatory T cells (Tregs) and ageing. This also clarifies why only a few smokers develop COPD (Figure 2).

## 5. Conclusions

In conclusion, current research suggests that Tregs are pivotal in the pathogenesis of COPD by preserving peripheral immune tolerance. Various clinical studies have shown reduced levels of Tregs in the lung tissue, bronchoalveolar lavage fluid, and peripheral blood of COPD patients compared to both healthy subjects and smokers without COPD. Methodological differences in Treg identification and sample analysis may lead to conflicting results. However, most studies assume a protective function of Tregs, and their deficiency is associated with the development of COPD or more severe stages of the disease. Similar to the progression of malignancies, inflammatory diseases, and autoimmune disorders, pulmonary ageing is associated with the decline in Treg functionality. This topic will continue to be an area of research interest in the upcoming years as further exploration is warranted.

## Figures and Tables

**Figure 1 ijms-26-03721-f001:**
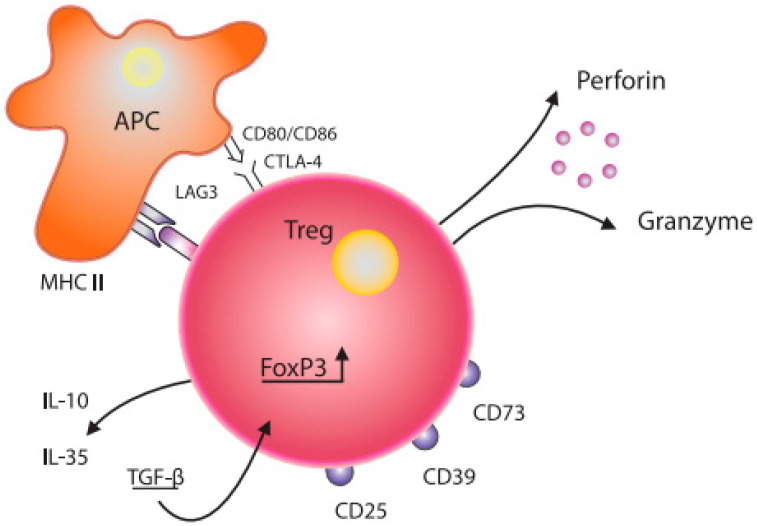
Schematic diagram of the structure of a Treg cell and its interaction with a dendritic cell. Treg—T regulatory cell; FoxP3—transcription factor forkhead-box-P3; APC—antigen-presenting cell; CD80/CD86, CTLA-4—co-stimulation molecules; IL-10, IL—35, TGF-β—inhibitory cytokines; CD25—IL-2 receptor; CD73, CD39, LAG-3—inhibitory molecules; MHC II—major histocompatibility complex class II.

**Figure 2 ijms-26-03721-f002:**
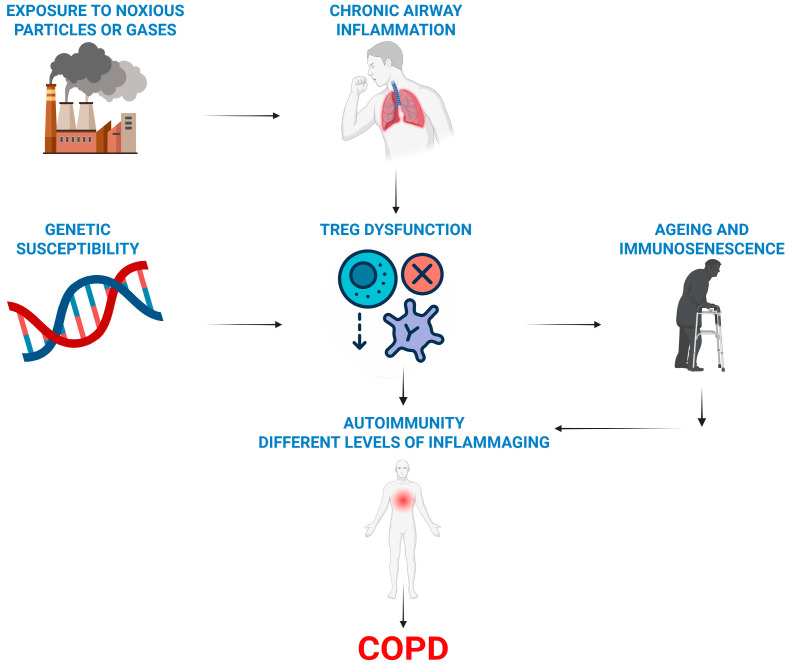
Overview of the immune hypothesis of COPD. Treg dysfunction (a lack or their impaired activity) combined with a genetic predisposition promotes autoimmune and ageing processes in smoking-damaged lungs and causes COPD.

**Table 1 ijms-26-03721-t001:** Summary of the main findings regarding Treg and COPD in clinical studies [49].

	Pulmonary Response	Systemic Response
**Treg**	↓ lung tissue of COPD (II-IV) vs. control [50] ↑ lung tissue of COPD (I-II) vs. healthy smokers [51] ↓ BALF of COPD (II-III) vs. healthy smokers [37] ↓ lung tissue of COPD (II-III) vs. healthy smokers and control [31,52] ↓ bronchial epithelium of COPD (III-IV) vs. COPD (I-II) [30] ↓ BALF of COPD (rapid decline in lung function) vs. COPD (nonrapid decline) [32] ↓ small airways of COPD (I-IIII) vs. control [42,43] ↑ large airways of COPD (II) vs. control [43] ↑ BALF of COPD (I-III) and healthy smokers vs. control [38] ↑ lymphoid tissue of COPD (I-III) vs. control [42,46] ↓lung tissue of COPD vs. healthy smokers [53]	↓ peripheral blood of COPD (II-III) vs. healthy smokers and control [54] ↓ peripheral blood of COPD (I-II) vs. healthy smokers [51] ↑ peripheral blood of COPD (II-IV) vs. control [55] ↓ peripheral blood of COPD (III-IV) vs. COPD (I-II) [30] ↓ peripheral blood of exacerbated COPD vs. stable COPD [33] ↓ peripheral blood of exacerbated COPD vs. stable COPD, healthy smokers, and control [52] ↓ peripheral blood of COPD (I-II) vs. healthy smokers and control [29]

Data are presented and divided by the evaluated compartment (pulmonary and systemic response). Treg, regulatory T cell; COPD, chronic obstructive pulmonary disease; vs., versus; BALF, bronchoalveolar lavage fluid; ↑: increased; ↓: decreased.

## Data Availability

Not applicable.

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
