# Peer review of "Role of Regulatory T Cells in Pulmonary Ageing and COPD Development"

_ijms, 2025, doi:10.3390/ijms26083721_

Round 1

Reviewer 1 Report

Comments and Suggestions for Authors

General Comments
This is a interesting,  topical and generally comprehensive review of an important subject area.  

Major Comments
1. Some of the references are quite dated and, unless they are clearly classic primary articles that deserve continued citations, those earlier than, say 2010, should be updated with more recent articles.  

2.  Sections presenting complex, and in some cases conflicting data (e.g., 3.1, 3.5), would benefit from summary sentences. 

3. (lines 71-74). The sentence “Effector Treg cells 71 adopt tissue-specific transcription factors, such as T-bet, Gata3, and RORγt, acquiring specialised regulatory functions and functioning as tissue-resident Treg cells that uphold immune balance within parenchymal tissues “ is problematic for several reasons.  First, the cited transcription factors are not tissue-specific, but instead specify different T-effector cell phenotypes (commonly known as T1, T2, and T17, respectively).  Second, references #5-9 do not support this statement.  Third, although Tregs can convert into T17 effectors, which is touched upon in lines 108-11; that process is entirely different from what is being described starting at line 71.  Please revise.  

Minor Comments
1. The airflow obstruction in COPD can better be described as “incompletely” or “not-fully reversible” (e.g., GOLD 2024 Scientific report), rather than irreversible.  

(lines 75-85): to avoid potential confusion, “GARP” should be identified as also being know as LRRC32 (leucine rich repeat containing 32; Gene ID 2615 in humans)

(lines 190-192 and Table 1): decreased expression of FoxP3+ cells in the lungs in COPD was also demonstrated in two additional papers, PMIDs: 24805101, 20368170 )

(lines 235-236) The impact of leptin on Tregs is complex and there are conflicting results in different diseases and in differing murine disease models.  The reference most relevant to this review articles is that of Bruzzantiti et al. (PMID: 31308239), which showed that increased leptin levels in COPD are linked to inhibition of PoxP3 expression.  Those data contradict the overall thesis of this section.  Please expand on this topic to fully analyze the current literature, and ideally suggest experiments to resolve the question.   

(lines 259-270): although this description of the paper by Kim and Sin accurately conveys the argument of those authors, that recent paper is highly speculative and presents no direct evidence to support their thesis.  Would suggest either dropping it, or presenting a more critical analysis of that paper.  

Author Response

Major Comments

  1. Some of the references are quite dated and, unless they are clearly classic primary articles that deserve continued citations, those earlier than, say 2010, should be updated with more recent articles.  Answer: 

    Dear Rewiever, we sincerely thank you for your comments and valuable advice. 

    We agree that the cited sources should be as recent as possible. But in our opinion, all the works we cite, even if not so new, are interesting and significant. We hope that soon, more studies will be conducted on the influence of Tregs on the development of COPD.
  2. Sections presenting complex, and in some cases conflicting data (e.g., 3.1, 3.5), would benefit from summary sentences. Answer: Thank you for pointing out that summary sentences are needed for sections 3.1 and 3.5. We have tried to do this.

  3.  (lines 71-74). The sentence “Effector Treg cells 71 adopt tissue-specific transcription factors, such as T-bet, Gata3, and RORγt, acquiring specialised regulatory functions and functioning as tissue-resident Treg cells that uphold immune balance within parenchymal tissues “ is problematic for several reasons.  First, the cited transcription factors are not tissue-specific, but instead specify different T-effector cell phenotypes (commonly known as T1, T2, and T17, respectively).  Second, references #5-9 do not support this statement.  Third, although Tregs can convert into T17 effectors, which is touched upon in lines 108-11; that process is entirely different from what is being described starting at line 71.  Please revise.  Answer: 

    1. Based on your comment, we reviewed the paragraph and decided to remove it.

       Minor Comments

      1. The airflow obstruction in COPD can better be described as “incompletely” or “not-fully reversible” (e.g., GOLD 2024 Scientific report), rather than irreversible.  Answer: 

      Based on your recommendation, we changed the word "irreversible" to "not fully completed".

      2. (lines 75-85): to avoid potential confusion, “GARP” should be identified as also being know as LRRC32 (leucine rich repeat containing 32; Gene ID 2615 in humans). Answer: Done. 

      3. (lines 190-192 and Table 1): decreased expression of FoxP3+ cells in the lungs in COPD was also demonstrated in two additional papers, PMIDs: 24805101, 20368170 )

      Answer: We have improved Table 1 with the source you provided. However, the second article (PMDI 20368170) is about emphysema, and emphysema does not always cause COPD, so we believe that this source should not be cited in the context of our topic.

      4. (lines 235-236) The impact of leptin on Tregs is complex and there are conflicting results in different diseases and in differing murine disease models.  The reference most relevant to this review articles is that of Bruzzantiti et al. (PMID: 31308239), which showed that increased leptin levels in COPD are linked to inhibition of PoxP3 expression.  Those data contradict the overall thesis of this section.  Please expand on this topic to fully analyze the current literature, and ideally suggest experiments to resolve the question.   Answer: 

      Thank you for your comments on the importance of leptin. However, as you mentioned, this is an extensive topic that deserves a separate review article, so we have only added a little to this section.

      5. (lines 259-270): although this description of the paper by Kim and Sin accurately conveys the argument of those authors, that recent paper is highly speculative and presents no direct evidence to support their thesis.  Would suggest either dropping it, or presenting a more critical analysis of that paper.  Answer:

      We fully agree with your opinion of the Kim and Sin article and have critically evaluated it and noted this in the text.

Reviewer 2 Report

Comments and Suggestions for Authors

The authors present a review summarizing the scientific evidence of Treg cells' role in COPD. Interestingly, the researchers agree entirely on whether these cells' role is protective or inducer. The quantity and quality of the information presented in this review are good enough. However, some aspects of the work need improvement.

  1. Oxidative stress (OS) is the primary causative agent of COPD, and several studies have shown that OS modulates the function of Treg cells. It would be interesting to add a paragraph indicating these aspects between Treg cells modulating by OS in the context of COPD.
  2. The work lacks many figures and schemes, especially in the first issues of the review. I suggest adding complementary figures that help review readers to have a better experience with the lecture. For example, parts 2 and 3 could be summarized in schematic figures containing artwork relative to airway cells and Treg cells.
  3. Figure 1 could be improved by adding artwork, such as a drawing of a cell or airway epithelium affected by aging, for example.
  4. A minor mistake is observed in lane 41 with the word INTRINCATE. This word must be corrected to INTRINCATED.

Author Response

  • Oxidative stress (OS) is the primary causative agent of COPD, and several studies have shown that OS modulates the function of Treg cells. It would be interesting to add a paragraph indicating these aspects between Treg cells modulating by OS in the context of COPD. Response: 

    Dear Reviewer,

    We sincerely thank you for your comments and valuable advice.

    Of course, your comment about the role of oxidative stress in the pathogenesis of COPD is valuable. Therefore, we have included statements about OS in the introduction.

  • The work lacks many figures and schemes, especially in the first issues of the review. I suggest adding complementary figures that help review readers to have a better experience with the lecture. For example, parts 2 and 3 could be summarized in schematic figures containing artwork relative to airway cells and Treg cells. Response: 

    Indeed, we agree that a visual illustration of the text was missing, so we created and added a diagram in Fig.1.

  • Figure 1 could be improved by adding artwork, such as a drawing of a cell or airway epithelium affected by aging, for example. Response: 

    Taking your advice into account, we have improved the scheme (now Fig.2)

  • A minor mistake is observed in lane 41 with the word INTRINCATE. This word must be corrected to INTRINCATED. Response: 

    The word INTRICATE was corrected into INTRICATED.
